# Where Do We Stand with Immunotherapy for Advanced Pancreatic Ductal Adenocarcinoma: A Synopsis of Clinical Outcomes

**DOI:** 10.3390/biomedicines10123196

**Published:** 2022-12-09

**Authors:** Liia Akhuba, Zhanna Tigai, Dmitrii Shek

**Affiliations:** 1Blacktown Mt Druitt Hospital, Sydney, NSW 2148, Australia; 2School of Health Sciences, Western Sydney University, Sydney, NSW 2150, Australia; 3Accreditation Centre, RUDN University, Moscow 117198, Russia; 4Blacktown Clinical School, Western Sydney University, Sydney, NSW 2148, Australia; 5Westmead Institute for Medical Research, Sydney, NSW 2145, Australia

**Keywords:** pancreatic cancer, immunotherapy, immune-checkpoint inhibitors, CAR T-cell therapy, cancer vaccines

## Abstract

Pancreatic cancer is the seventh leading cause of cancer-related mortality in both sexes across the globe. It is associated with extremely poor prognosis and remains a critical burden worldwide due to its low survival rates. Histologically, pancreatic ductal adenocarcinoma (PDAC) accounts for 80% of all pancreatic cancers; the majority of which are diagnosed at advanced stages, which makes them ineligible for curative surgery. Conventional chemotherapy provides a five-year overall survival rate of less than 8% forcing scientists and clinicians to search for better treatment strategies. Recent discoveries in cancer immunology have resulted in the incorporation of immunotherapeutic strategies for cancer treatment. Particularly, immune-checkpoint inhibitors, adoptive cell therapies and cancer vaccines have already shifted guidelines for some malignancies, although their efficacy in PDAC has yet to be elucidated. In this review, we summarize the existing clinical data on immunotherapy clinical outcomes in patients with advanced or metastatic PDAC.

## 1. Introduction

Pancreatic cancer is the seventh leading cause of cancer-related mortality in both sexes across the globe [1]. It remains a critical burden worldwide due to its low survival rates and extremely aggressive nature [2]. A total of 95% of pancreatic malignancies arise from exocrine parts (ductal epithelium, acinar cells and connective tissue), and another 5% develop from endocrine parenchyma [3]. Histologically, pancreatic ductal adenocarcinoma (PDAC) accounts for 80% of all pancreatic cancers [4], and the majority of cases are diagnosed at advanced stages (Figure 1) [5]. Localized cases can be treated with surgery, however, the five-year overall survival (OS) rate does not exceed 25% [6]. Unfortunately, there are no curative strategies for advanced stages, and palliative chemoradiotherapy reaches a five-year OS rate of less than 8% [7]. Chemotherapeutic choices of treatment include either modified FOLFIRINOX (5-fluorouracil, leucovorin, irinotecan, oxaliplatin) [8], gemcitabine monotherapy or gemcitabine in combination with nab-paclitaxel [9].

Recent discoveries in cancer immunology resulted in the incorporation of immunotherapeutic strategies for the treatment of various solid and hematologic malignancies [10]. Multiple clinical trials showed the higher efficacy of immune-checkpoint inhibitors (ICIs) for melanoma, lung cancer, renal cell carcinoma, colorectal cancer and hepatocellular carcinoma [11,12,13,14,15]. Adoptive chimeric antigen receptor (CAR) T-cell therapy is another immunotherapeutic tool being established as effective for hematologic malignancies, particularly relapsed/refractory B cell lymphoma [16] or mantle-cell lymphoma [17]. Unfortunately, early-phase clinical trials have shown limited responses to immunotherapy in patients with PDAC [18,19]. Nevertheless, immunotherapy is still deemed the most likely way to improve prognosis for patients with advanced PDAC (aPDAC) in the near future [20]. In this review, we provide an overview of the clinical outcomes of immunotherapy in patients with advanced and/or metastatic PDAC.

**Figure 1 biomedicines-10-03196-f001:**
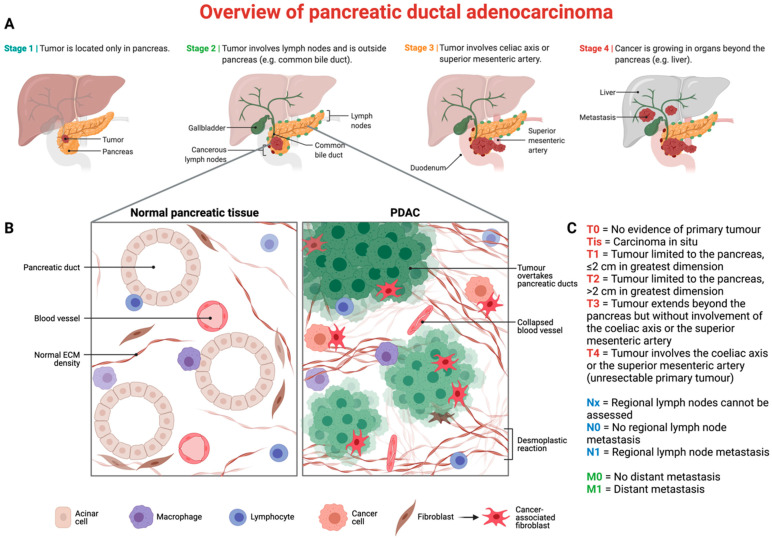
Overview of pancreatic ductal adenocarcinoma (PDAC): tumor microenvironment and classification. (**A**) Stages of PDAC’s progression, according to the AJCC’s 8th edition [21]. (**B**) Morphological differences between normal (**left**) and malignant (**right**) pancreatic tissue. (**C**) TNM classification of PDAC.

## 2. Immune-Checkpoint Inhibitors

ICIs have heralded a new era in clinical oncology [22]. ICIs are monoclonal antibodies that target immune checkpoints (e.g., cytotoxic T-lymphocyte antigen-associated protein 4 [CTLA-4], programmed cell death-1 [PD-1], programmed cell death ligand 1 [PD-L1]) on T-, cancer and antigen-presenting cells (Figure 2) [23]. The inhibition of immune checkpoints results in proper stimulation of T-cell receptor (TCR) signaling, differentiation of cytotoxic T cells and further destruction of malignant cells [24].

### 2.1. Blockade of Cytotoxic T-Lymphocyte-Associated Antigen 4

In 2011, Ipilimumab (IPI), an anti-CTLA-4 monoclonal antibody, became the first ICI to be approved by the US Food and Drug Administration (FDA) [25]. Though it shifted treatment guidelines for melanoma [26], trials elucidating IPI in patients with aPDAC did not show any promising results. In particular, a single-arm phase 2 trial established that IPI (3 mg/kg) monotherapy in aPDAC patients (74% [*n* = 20] pre-treated with gemcitabine-based regimens) failed to show any durable response [27]. Furthermore, 11.1% (*n* = 3) of patients developed severe (≥grade 3) immune-related adverse events (irAEs) [27].

A dose-escalation phase 1b trial (NCT01473940) presented that IPI in combination with gemcitabine reached an objective response rate (ORR) of 14% (*n* = 3) [28]. Moreover, the median progression-free survival (PFS) was 2.78 months (95% CI, 1.61 to 4.83) with a median OS of 6.9 months (95% CI, 2.63 to 9.57) [28]. A total of 19% (*n* = 4) developed irAEs of grade 3 or higher [28]. Overall, a combinatorial therapy of gemcitabine 1000 mg/m^2^ + IPI 3 mg/kg was considered safe; however, the efficacy outcomes did not exceed the results of gemcitabine monotherapy, and thus further exploration was halted.

Another CTLA-4 inhibitor tremelimumab was studied in a phase 2 basket trial (NCT02527434) [29]. A pooled analysis of patients with aPDAC (pre-treated with gemcitabine) revealed that 90% (*n* = 18) had disease progression [29]. The median OS reached 4 months (95% CI: 2.8 to 5.4) with 70% (*n* = 14) experiencing irAEs of grade 3 and higher. Thus, second-line tremelimumab monotherapy failed to show superior efficacy in patients with aPDAC [29].

Similar to IPI, a dose-escalation phase 1b study revealed that the combination of tremelimumab with gemcitabine showed better effectiveness than monotherapy [30]. In particular, the median OS reached 7.4 months (95% CI, 5.8 to 9.4) with 6% (*n* = 2) reaching PR [30] and only 3% (*n* = 1) developing serious irAEs [30].

A phase 2 trial (NCT02558894) established that a combination of tremelimumab 75 mg + durvalumab 1500 mg (anti-PD-L1 inhibitor) reached an ORR of 3.1% (*n* = 1) with 22% (*n* = 7) developing irAEs of grade 3 and higher [31]. The trial failed to reach the pre-designed ORR threshold of 10%, and further exploration was stopped.

Finally, recent results of a randomized phase 2 trial (NCT02879318) comparing arm A (gemcitabine + nab-paclitaxel) to arm B (gemcitabine + nab-paclitaxel + durvalumab + tremelimumab) established no significant difference in survival [32]. In particular, the median OS was 8.8 months (95% CI: 7.2 to 11.2) and 9.8 months (95% CI: 8.3 to 12.2) in arms A and B, respectively (Hazard Ratio [HR]: 0.94, *p* = 0.72) [32]. The median PFS was 5.4 months (95% CI: 3.6 to 6.6) and 5.5 months (95% CI: 3.8 to 5.7) in arms A and B, respectively (HR: 0.98, *p* = 0.91) [33]. Serious adverse events were reported in 44.8% (*n* = 26) and in 68.9% (*n* = 82) of patients in arms A and B, respectively [33]. Thus, combinatorial immunotherapy failed to show a significantly higher efficacy in aPDAC patients than the current standard of care (SoC).

To sum up, the existing results do not show encouraging outcomes for CTLA-4 inhibitors in patients with aPDAC. Strikingly, even the combination of the PD-L1 inhibitor durvalumab + CTLA-4 inhibitor tremelimumab is not significantly better than SoC chemotherapy. This data emphasizes the need for further studies on novel combinatorial approaches able to overcome the unique immunosuppressive and fibrotic morphological features of PDAC that potentially play a major role in the inhibition of existing ICI-based regimens. Moreover, further trials should be designed for the head-to-head comparison of a single ICI or a combination of CTLA-4 + PD-1 inhibitors to the current SoC or SoC + ICI.

### 2.2. Blockade of Programmed Cell Death-1 with Its Ligands

Pembrolizumab (PEMBRO) and nivolumab (NIVO) are humanized monoclonal antibodies (mAbs) inhibiting the PD-1 inhibitory checkpoint [34]. They have been granted FDA approval for the treatment of melanoma, lung cancer and aPDAC with MSI-H status [19].

A single-arm phase 2 trial (NCT01876511) reported that cohort C (non-colorectal cancer patients [*n* = 7 out of whom *n* = 4 with PDAC] with a mismatch repair deficiency [dMMR] or a high microsatellite instability [MSI-H]) treated with PEMBRO as a ≥2 line of therapy reached an ORR of 71% with a median PFS of 5.4 months [35]. The authors concluded that PEMBRO alone showed a durable clinical efficacy in the MSI-H patients from cohort C [36]. Later reports from the NCT01876511 trial also observed durable response in the cohort of patients (*n* = 86) with PDAC (ORR 53% [95% CI: 42% to 64%]; 21% of patients reached complete response [CR]) [37]. A recent report from the multicenter KEYNOTE-158 trial reported that out of *n* = 22 patients only *n* = 4 (18.2%) had either CR or a partial response [PR] [38]. Nonetheless, the prevalence of PDAC patients with MSI-H is between 0.8% and 2% [36], which stresses the extreme necessity for searching for other therapeutic strategies effective for the majority of PDAC patients.

A single-arm phase 2 study (JapicCTI-184230) elucidated the efficacy of NIVO (480 mg every 4 weeks) in combination with modified FOLFIRINOX (oxaliplatin 85 mg/m^2^, levofolinate 200 mg/m^2^, irinotecan 150 mg/m^2^ and fluorouracil 2400 mg/m^2^ every 2 weeks) in treatment-naïve patients with metastatic PDAC [39]. The authors reported that the ORR reached 32.3% (only PR) with a median OS and PFS of 13.4 and 7.39 months, respectively [39]. Moreover, 54.8% (*n* = 17) of patients reached the one-year survival threshold. Further validation of this regimen is currently ongoing.

A randomized multi-cohort phase 2 trial (NCT03214250) established that first-line NIVO in combination with gemcitabine and nab-paclitaxel reached a primary efficacy endpoint in *n* = 34 aPDAC patients [40]. ORR reached 50% (95%CI: 32 to 68) with a median OS of 16.7 months (95% CI: 9.8 to 18.4) [40]. A total of 57.7% of patients surpassed the one-year threshold disregarding their mutational status. In contrast, Wainberg et al. reported that NIVO in combination with gemcitabine and nab-paclitaxel showed only 18% (95% CI: 8.6 to 31.4) as a first-line therapy for patients with advanced/metastatic PDAC [41]. Moreover, *n* = 48 (96%) experienced grade 3–4 immune-related adverse events (irAEs) [41]. The concerning safety profile of this regimen did not support further investigation.

Finally, a randomized phase 2 CheckPAC study (NCT02866383) established that NIVO in combination with IPI and stereotactic body radiotherapy (SBRT) of 15 Gy reached a 37.2% ORR in patients with metastatic PDAC [42]. The authors concluded that the studied regimen showed a favorable efficacy, and further studies are currently ongoing. Nevertheless, the role of SBRT is unknown.

In summary, the current evidence supports that PEMBRO is effective for aPDAC patients with MSI-H status, although the data were obtained from basket single-arm trials primarily powered by elucidating PEMBRO’s efficacy in colorectal cancer patients. NIVO in combination with (1) chemotherapy or (2) IPI + SBRT showed meaningful response rates; however, the head-to-head comparison with current SoC chemotherapy regimens is necessary to precisely elucidate the safety and efficacy profiles of this regimen.

A dose-escalating phase 1b trial (NCT02323191) determined the role of the PD-L1 inhibitor Atezolizumab (Atezo) in combination with Emactuzumab, an inhibitor of the colony-stimulating factor-1 receptor (CSF1R) in patients with solid tumors [43]. Gomez-Roca et al. reported that the ORRs in treatment-naïve aPDAC patients reached 9.8% and 12.5% for patients previously treated with other ICI-based regimens, albeit a separate analysis of patients with only aPDAC was not conducted [43]. Further clinical studies were not warranted due to the full potential of CSF1R being unknown and the high rate of grade 3–4 irAEs.

Another PD-L1 inhibitor, Avelumab, in combination with Binimetinib (MEK [mitogen-activated protein kinase] inhibitor) was determined in a dose-escalating phase 1b trial (NCT03637491) [44]. The ORR reached 0% (95% CI: 0 to 30.8) and 8.3% (95% CI: 0.2 to 38.5) in metastatic PDAC patients from cohorts A (binimetinib 30 mg) and B (binimetinib 45 mg), respectively [44]. *n* = 12 (54.5%) experienced irAEs of grade 3 and higher [44]. The study was terminated before the optimal dose for phase 2 was established.

Finally, despite ICIs having demonstrated their superior efficacy in aPDAC with MSI-H status, most phase 1 and 2 clinical trials failed to demonstrate any superior clinical efficacy in a majority of aPDAC patients as compared to the current SoC. Perhaps further data generated from ongoing clinical trials will reveal encouraging results for a combination of ICIs + radiotherapy and/or chemotherapy (Table 1). Nonetheless, further studies are critically needed to determine the mechanisms of PDAC’s resistance to ICI therapy.

## 3. Adoptive CAR T-Cell Therapy

Adoptive CAR (chimeric antigen receptor) T-cell therapy is a technology of the ex vivo expansion of a patient’s own T cells that have been genetically engineered to express CAR that recognizes a particular tumor antigen [45]. To date, no CAR T-cell therapy has been approved for the treatment of solid malignancies [45]. Regarding PDAC, a few clinical trials have demonstrated limited efficacy.

A single-arm phase 1 study (NCT01869166) determined the safety and efficacy of anti-EGFR (epidermal growth factor receptor) CAR T cells [46]. Among *n* = 14 pre-treated metastatic PDAC patients, the ORR reached 28.6% with a median PFS and OS of 3 and 4.9 months, respectively [46]. Each patient experienced a treatment-related adverse event (TRAE) of different severity [46]. Further trials were not supported due to the limited efficacy and the fact that EGFR is expressed by a wide range of tissues, which may explain the high incidence of TRAEs.

Another single-arm phase 1 basket trial (NCT01935843) elucidated the safety and efficacy of anti-HER-2 (human epidermal growth factor receptor 2). The median PFS reached 4.8 months in *n* = 11 aPDAC patients with more than 50% of HER-2 positive tumor cells. *n* = 1 (9%) reached PR, and *n* = 5 (45%) achieved stable disease (SD) [47]. No instances of high-grade lymphocytopenia or cytokine release syndrome were detected, and most low-grade toxicities were reversible [47]. The therapy showed encouraging clinical activity, and further trials are ongoing.

Based on promising preclinical mouse studies, two trials elucidated the clinical outcomes of anti-mesothelin CAR T cell therapy. A phase 1 trial (NCT01897415) showed that among *n* = 6 patients with aPDAC, none experienced severe TRAEs [48]. *n* = 2 (33.3%) reached SD [48]. Another phase 1 basket trial (NCT02159716) showed encouraging safety profiles in patients with chemo-refractory solid tumors [49]. *n* = 11 (73.3%) of patients achieved SD, and additional larger trials are currently ongoing.

Finally, a phase 1 basket trial (NCT02541370) elucidated the role of anti-CD133 CAR T-cell therapy [50]. Wang et al. established that within the cohort of patients with aPDAC pre-treated with cyclophosphamide/nab-paclitaxel, *n* = 3 (42%) reached SD, *n* = 2 (29%) PR and *n* = 2 (29%) had DP [50].

To date, knowledge regarding the clinical efficacy of CAR therapy in patients with aPDAC is very limited, although it remains a viable and promising topic of pancreatic cancer research. The identification of an ideal target is a major challenge for adoptive cell therapy in PDAC. Secondly, PDAC’s immune microenvironment comprised of macrophages, cancer-associated fibroblasts, myeloid-derived suppressor cells, dendritic cells, B cells and infiltrating regulatory T cells [51]. These cells can create a physical barrier for trafficking T cells and suppress T cell activation, resulting in a diminished efficacy of CAR T-cell therapy [52]. Overcoming those hurdles may one day result in successful implications of adoptive cell therapies for advanced and/or metastatic PDAC.

## 4. Cancer Vaccines

Cancer vaccines represent another promising strategy for PDAC treatment [53]. Vaccines can boost anti-tumor immunity by transferring the tumor antigens (Figure 3) in the form of cells, proteins or nucleic acids [53]. A number of clinical trials have elucidated the clinical efficacy of vaccines in patients with severely progressed PDAC.

### 4.1. Cell-Based Vaccines

A phase 2 open-label randomized trial (NCT02243371) determined the efficacy of the GVAX and CRS-207 vaccines with or without NIVO in patients with metastatic PDAC that progressed on one prior chemotherapy regimen [54]. GVAX is a cell-based vaccine that transfers an allogeneic tumor cell, engineered to express granulocyte-macrophage colony-stimulating factor (GM-CSF) [55]. CRS-207 is a microorganism-based vaccine that transfers a live-attenuated *Listeria monocytogenes* (*Lm*) engineered to express the PDAC-associated antigen mesothelin [56]. Recruited patients received Cyclophosphamide + GVAX + CSR-207 with (cohort A, *n* = 51) or without (cohort B, *n* = 42) NIVO [54]. The median OS reached 5.9 (95% CI: 4.7 to 8.6) and 6.1 (95% CI: 3.5 to 7) months in cohorts A and B, respectively [54]. The ORR was 4% and 2% in cohorts A and B, respectively [54]. The trial failed to meet its primary endpoint of OS improvement, and durable response rates were not registered.

Another phase 2 (NCT02004262) randomized trial (ECLIPSE study) compared the efficacy of Cyclophosphamide + GVAX + CRS-207 (arm A) and CRS-207 (arm B) to a physician’s choice of an SoC chemotherapy [57] in patients with metastatic PDAC who previously failed on >2 lines of chemotherapy. The median OS reached 3.7 (95% CI: 2.9 to 5.3), 5.4 (95% CI: 4.2 to 6.4) and 4.6 (95% CI: 4.2 to 5.7) months in arms A, B and C, respectively [57]. No significant difference as compared to control arm C was registered. The authors concluded that the combination of GVAX + CRS-207 failed to show higher efficacy as compared to SoC chemotherapy in patients with metastatic PDAC [57].

A phase 3 randomized PILLAR trial (NCT01836432) determined the clinical role of Algenpantucel-L + FOLFIRINOX (arm A, *n* = 145), compared to FOLFIRINOX (arm B, *n* = 158) alone [58]. Algenpantucel-L is a whole-cell vaccine that consists of two irradiated allogeneic pancreatic cancer cell lines (HAPa-1, HAPa-2) engineered to express the murine enzyme α (1,3)-galactosyltransferase (αGT) responsible for hyperacute rejection [59]. The study results revealed that the median OS was 14.3 (95% CI: 12.6 to 16.3) and 14.9 (95% CI: 12.2 to 17.8) months in arms A and B, respectively [58]. Moreover, the median PFS was 12.4 and 13.4 months in arms A and B, respectively (Hazard Ratio [HR]: 1.33 [95% CI: 0.66 to 1.58], *p* = 0.59) [58]. Thus, Algenpantucel-L failed to show improved survival in patients with local aPDAC compared to FOLFIRINOX [58].

Dendritic cells (DCs) are antigen-presenting cells and play a crucial role in the anti-tumor immune response. A phase 1 single-arm trial (NCT01410968) determined the role of DCs isolated from the peripheral blood of PDAC patients with HLA-A2 positive status. DCs were pulsed with three distinct A2-restricted peptides: (1) human telomerase reverse transcriptase (hTERT, TERT572Y); (2) carcinoembryonic antigen (CEA, Cap1-6D) and (3) surviving (SRV.A2) [60]. The median OS reached 7.7 months, and *n* = 4 (33.3 %) reached SD [60]. The treatment was well tolerated, and the flow cytometry analysis revealed that patients with SD had a high expansion of antigen-specific T cells [60]. This method was considered promising because DCs can be pulsed with other synthetic peptides, and further larger trials were recommended.

### 4.2. Peptide-Based Vaccines

The vaccine KIF20A-66 is a human leukocyte antigen (HLA)-A24-restricted cytotoxic T cell epitope derived from KIF20A, a member of the kinesin superfamily protein 20A that is markedly upregulated in PDAC [61]. A phase 1/2 single-arm trial (UMIN000004919) reported that vaccinated gemcitabine-pre-treated patients with metastatic PDAC had a median OS and PFS of 4.7 and 1.9 months, respectively [62]. *n* = 21 (72%) of patients reached SD [62]. Asahara et al. concluded that KIF20A-66 showed higher survival rates in patients with metastatic PDAC as compared to the best supportive care, and further trials were encouraged.

A phase 2 single-arm non-randomized VENUS-PC trial (UMIN000008082) determined the role of HLA-A*2402-restricted KIF20A-derived peptide vaccine in combination with the gemcitabine and antiangiogenic vaccines targeting vascular endothelial growth factor receptor 1 and 2 (VEGFR1, 2) [63]. The median OS reached 9 months with an ORR of 10.8% in *n* = 38 patients who had at least one allele of HLA-A*2402 (matched).

Finally, commonly tested KRAS and telomerase (GV1001)-targeting vaccines failed to show durable responses and/or superiority to gemcitabine in phase 2/3 clinical trials [64]. A phase 3 randomized TeloVac trial showed no significant difference in the OS for treatment-naïve patients with metastatic PDAC treated with chemotherapy or the GV1001 +/− chemotherapy vaccine [65].

Most of the trials determining the role of vaccines in patients with advanced PDAC failed to show durable response. Perhaps a critical factor responsible for such failure is the tumor microenvironment, which is characterized by an abundance of mesenchymal origin fibroblasts, blood vessels and tumor-infiltrating immune cells surrounded by extracellular matrix [66]. Those factors can inhibit the immune response, thus facilitating cancer escape from immunosurveillance [66]. In addition, vaccine therapy is challenged by a complex process of vaccine synthesis and the absence of a validated method to identify and/or measure the immune response to the vaccine. Nevertheless, further assessment in larger trials is necessary, especially in combination with other therapeutic strategies (Table 2).

## 5. Conclusions

PDAC is associated with an extremely poor survival rate and prognosis if diagnosed at late stages. To date, immunotherapy represents the biggest hope for improving clinical outcomes for patients with advanced/metastatic PDAC. Although PEMBRO has been approved for the treatment of aPDAC patients with MSI-H status, PDAC has demonstrated remarkable resistance to immunotherapy in the majority of cases. Further trials are extremely necessary to determine the role of combination approaches utilizing various immunotherapeutic strategies. Importantly, further trials should focus on overcoming therapeutic resistance by targeting multiple immune defects with several immunotherapeutic arms. Early trials have already reported the synergistic effect of ICIs or CAR therapies with chemoradiotherapy, albeit safety profiles should be closely monitored. In addition, future studies should prioritize integrated or convergent targets that can reprogram the tumor microenvironment rather than focusing on the depletion of a single/particular target. Another important point to consider is that an immunotherapeutic strategy should be based on an individual characterization of the tumor microenvironment of each patient. Such approach can be achieved by deep profiling of the tumor tissue during the pre-treatment stage with high-throughput technologies. This will promote the personalization of therapy, thus increasing clinical outcomes in patients with advanced and metastatic PDAC. Overall, such integration may facilitate the establishment of effective therapeutic strategies for the majority of PDAC patients in the near future.

## Figures and Tables

**Figure 2 biomedicines-10-03196-f002:**
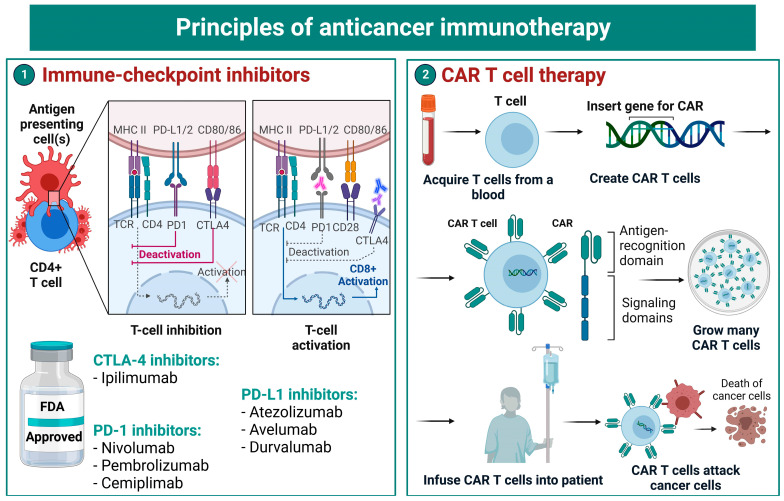
Overview of mechanisms of immune-checkpoint inhibitors and CAR T-cell therapy. MHC—major histocompatibility complex; PD-1—programmed cell death protein-1; PD-L1/2—programmed cell death ligand 1/2; CD—cluster of differentiation; TCR—T cell receptor; CTLA-4—cytotoxic T lymphocyte-associated antigen 4; FDA—US Food and Drug Administration; APC—antigen-presenting cell; IL—interleukin; IFN—interferon.

**Figure 3 biomedicines-10-03196-f003:**
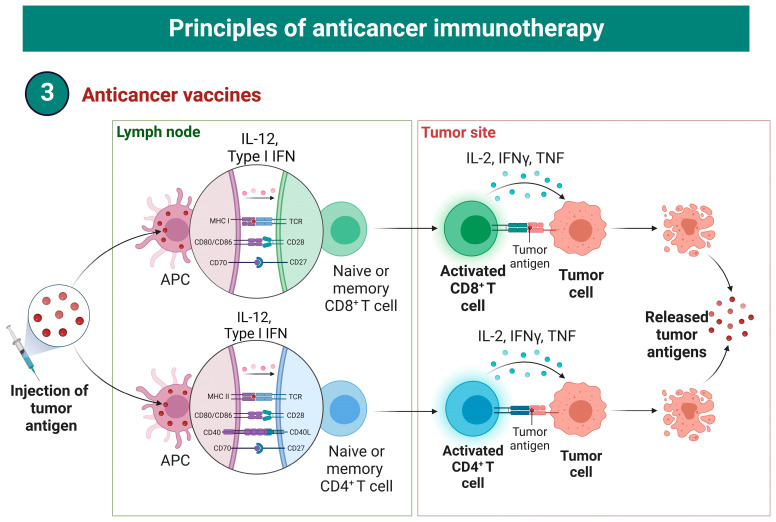
The overview of cancer vaccines’ mechanisms. MHC—major histocompatibility complex; CD—cluster of differentiation; APC—antigen-presenting cell; IL—interleukin; IFN—interferon.

**Table 1 biomedicines-10-03196-t001:** List of currently recruiting multicenter clinical trials determining immune-checkpoint inhibitors in patients with pancreatic cancer. Rand—randomization; NIVO—nivolumab; IPI—ipilimumab; PEMBRO—pembrolizumab; ATEZO—atezolizumab; TRAEs—treatment-related adverse events; DLTs—dose-limiting toxicities; PFS—progression-free survival; ORR—objective response rate; mAb—monoclonal antibody; CTLA-4—cytotoxic T-lymphocyte-associated antigen 4; PD-1—programmed cell death protein 1; SCCHN—squamous cell carcinoma of head and neck; RCC—renal cell carcinoma; DOR—duration of response; TTR—time to response; EGFR—epidermal growth factor receptor; TGFβ—transforming growth factor β.

NCT	Phase	Rand.	Sponsor	ICI	Endpoint(s)	Estimated Date for Primary Results
NCT03816358	1/2	No	National Cancer Institute	Arm I: Anetumab + NIVO. Arm II: Anetumab + IPI + NIVO. Arm III: Anetumab + Gemcitabine + NIVO.	Primary: maximum tolerated dose. Secondary: biomarker analysis.	January 2024
NCT04666688	1/2	No	PureTech	Dose-expansion arm: LYT-200 (Galectin-9 inhibitor) + PD-1 inhibitor + Gemcitabine + Nab-paclitaxel.	Primary: Incidence of TRAEs, incidence of DLTs, PFS and ORR. Secondary: Pharmacokinetics and pharmacodynamics of LYT-200.	December 2022
NCT04140526	1/2	No	National Cancer Institute; OncoC4 Inc.	Arm I: ONC-392 (CTLA-4 inhibitor). Arm II: ONC-392 + Pembrolizumab.	Primary: DLT, maximum tolerated dose, incidence of TRAEs. Secondary: ORR, PFS, OS.	December 2023
NCT04152018	1	No	Pfizer	Dose-escalation arm: PF-06940434 (PD-1 inhibitor). Dose-finding arm: PF-06940434 + PF-06801591 (PD-1 inhibitor). Dose-expansion form A: PF-06940434 + PF-06801591 in SCCHN. Dose-expansion form B: PF-06940434 + PF-06801591 in RCC.	Primary: DLT, PFS and incidence of TRAEs. Secondary: Pharmacokinetics and pharmacodynamics of PF-06940434 and PF-06801591, DOR.	September 2023
NCT04336098	1	No	Surface Oncology; MSD LLC	Arm III: SRF617 + PEMBRO. Arm IV: SRF617 + PEMBRO + Gemcitabine + albumin-bound Paclitaxel.	Primary: DLT Secondary: Pharmacokinetics and pharmacodynamics of SRF617, PFS.	November 2022
NCT04332653	1/2	No	NeoImmune Tech	NT-I7 (Efineptakin Alfa, long-acting human interleukin-7) + PEMBRO	Primary: Safety and tolerability of NT-I7 Secondary: DOR, PFS, OS, ORR, incidence of irAEs	May 2024
NCT05293496	1	No	MacroGenics	MGC018 (B7-H3 inhibitor) + Lorigerlimab (bispecific CTLA-4/PD-1 inhibitor)	Primary: incidence of irAEs Secondary: Pharmacokinetics and pharmacodynamics of MGC018, DOR, OS, PFS, ORR.	March 2024
NCT03915678	2	No	Institut Bergonie	ATEZO + BDB001 (Toll-like receptor 7/8 agonist) + Radiotherapy	Primary: assessment of antitumor activity Secondary: PFS, ORR	September 2023
NCT04548752	2	Yes	National Cancer Institute	Control arm: Olaparib. Experimental arm: Olaparib + PEMBRO.	Primary: PFS Secondary: Incidence of irAEs, OS, ORR, DOR	March 2025
NCT03485209	2	No	Seagen Inc.; MSD LLC	Arm IV: Tisotumab vedotin + PEMBRO + carboplatin + cisplatin.	Primary: ORR Secondary: incidence of irAEs, DOR, TTR, PFS, OS	November 2023
NCT04429542	1	No	Bicara Therapeutics	Arm II: BCA101 (EGFR and TGFβ fusion mAb + PEMBRO.	Primary: Safety and incidence of DLTs Secondary: ORR, PFS, DOR, OS	December 2023
NCT04561362	1/2	No	BicycleTx Limited	Dose-escalation cohort A2: BT8009 + NIVO. Dose-expansion B2: BT8009 + NIVO.	Primary: DLTs, ORR, PFS, OS Secondary: DOR	June 2023
NCT02834013	2	No	National Cancer Institute	Arm I: IPI + NIVO. Arm II: NIVO.	Primary: ORR Secondary: incidence of irAEs, OS, PFS	October 2023

**Table 2 biomedicines-10-03196-t002:** List of currently recruiting multicenter clinical trials determining clinical outcomes of adoptive T-cell therapies and vaccines in patients with pancreatic adenocarcinoma. Rand—randomization; CAR—chimeric antigen receptor; HER-2—human epidermal growth factor receptor 2; MUC1—mucin 1; MTD—maximum tolerated dose; DC—dendritic cell; DLTs—dose-limiting toxicities; ORR—objective response rate; TRAEs—treatment-related adverse events; OS—overall survival; CEA—carcinoembryonic antigen.

NCT	Phase	Rand.	Sponsor	Intervention	Primary Endpoint(s)	Estimated Date for Primary Results
Adoptive cell therapies						
NCT04404595	1/2	No	CARsgen Therapeutics Co., Ltd.	CAR T cells (Claudin 18.2)	Incidence of TRAEs, ORR	June 2025
NCT04660929	1	No	Carisma Therapeutics Inc	CAR macrophages (HER-2)	Incidence of TRAEs	February 2023
NCT05239143	1	No	Poseida Therapeutics, Inc.	CAR T cells (MUC1)	MTD; ORR and incidence of TRAEs	April 2026
NCT04157127	1	No	Baylor College of Medicine	Autologous DC vaccine	MTD, DLTs	January 2024
NCT04581473	1/2	Yes	CARsgen Therapeutics Co., Ltd.	Experimental arm: CT041 (CAR T cells [Claudin 18.2]). Control arm: Chemotherapy or PD-1 inhibitor.	Incidence of TRAEs, MTD, PFS	June 2024
NCT04348643	1/2	No	Chongqing Precision Biotech Co., Ltd.	CAR T cells (CEA)	Incidence of TRAEs	January 2023
Cancer vaccines						
NCT03323944	1	No	University of Pennsylvania	CAR T cells (Mesothelin)	Response rate, PFS, OS	September 2024
NCT03953235	1/2	No	Gritstone bio, Inc. and Bristol Myers Squibb	GRT-C903 + GRT-R904 + Nivolumab + Ipilimumab	Incidence of TRAEs, ORR	December 2023
NCT04807972	2	Yes	AbbVie	Control arm: FOLFIRINOX. Experimental arm I: FOLFIRINOX + ABBV-927 Experimental arm II: ABBV-927 + Budiglimab + mFOLFIRINOX.	OS	August 2024
NCT04853017	1	No	Elicio Therapeutics	ELI-002 (a lipid-conjugated immune-stimulatory oligonucleotide [Amph-CpG-7909] plus a mixture of lipid-conjugated peptide-based antigens [Amph-Peptides])	MTD, safety	November 2024
NCT02600949	1	No	M.D. Anderson Cancer Center	Arm I: personalized vaccine + imiquimod. Arm II: personalized vaccine + imiquimod + pembrolizumab. Arms III and IV: vaccine + imiquimod + pembrolizumab + APX005M.	Incidence of TRAEs	May 2025
NCT04111172	2	Yes	Thomas Jefferson University	Experimental arm I: adenovirus 5/F35-human guanylyl cyclase C-PADRE vaccine (low dose) Experimental arm II: medium dose Experimental arm III: high dose	Incidence of TRAEs, Antigen-specific T-cell response to guanylyl cyclase C (GCC)	March 2024
NCT02451982	2	Yes	Sidney Kimmel Comprehensive Cancer Center at Johns Hopkins and Bristol Myers Squibb	Experimental arm I: CY/GVAX alone. Experimental arm II: CY/GVAX with nivolumab. Experimental arm III: CY/GVAX with nivolumab and urelumab. Experimental arm IV: BMS-986253 and Nivolumab.	IL17A expression, Intratumoral CD8+ CD137+ cells, Intratumoral granzyme B PD-1+ CD137+ cells, Pathologic Response	June 2023
NCT03767582	1/2	Yes	Sidney Kimmel Comprehensive Cancer Center at Johns Hopkins and Bristol Myers Squibb	Arm I: Nivolumab/CCR2/CCR5 dual antagonist. Arm II: Nivolumab/GVAX/CCR2/ CCR5 dual antagonist	Percentage of participants who have >80% increase in infiltration of CD8+ CD137+ T cells into the PDAC after treatment compared to baseline	March 2023

## Data Availability

All data is available upon a request to a corresponding author.

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
