# Peer review of "Where Do We Stand with Immunotherapy for Advanced Pancreatic Ductal Adenocarcinoma: A Synopsis of Clinical Outcomes"

_biomedicines, 2022, doi:10.3390/biomedicines10123196_

Round 1

Reviewer 1 Report

This review  summarizes  existing clinical data of immunotherapy efficacy in patients with advanced or metastatic  PDAC.

 My comments.

 1. The abstract  could be more informative: it would be helpful to list the main immunotherapeutic strategies that are being considered for PDAC treatment.

2. From the review It remains unclear the clinical efficacy of immunotherapy as monotherapy, as well as combined immunotherapy (without immunosuppressive chemotherapy).

3. The “conclusions” section is  very short. In this section, the authors could more broadly outline  the prospects for the use of immunotherapy for  PDAC treatment 

Author Response

This review summarizes existing clinical data of immunotherapy efficacy in patients with advanced or metastatic PDAC.

The abstract could be more informative: it would be helpful to list the main immunotherapeutic strategies that are being considered for PDAC treatment.

Thank you for the comment. We have specified the immunotherapeutic arms currently used for PDAC treatment in the abstract.

From the review It remains unclear the clinical efficacy of immunotherapy as monotherapy, as well as combined immunotherapy (without immunosuppressive chemotherapy)

Thank you for this valuable comment. We have added some information regarding this topic in the resubmitted version of the manuscript. However, it should be said that unfortunately no head-to-head comparison of various IO regimens had been studied so far.

The “conclusions” section is very short. In this section, the authors could more broadly outline the prospects for the use of immunotherapy for PDAC treatment 

Thank you for the comment. In the resubmitted version we have more broadly discussed the future trajectory of IO in PDAC.

Reviewer 2 Report

The manuscript "Where do we stand with immunotherapy for advanced pancreatic ductal adenocarcinoma: a synopsis of clinical outcomes" is well written. I have only minor comments

-Starting from the title, what is " Ade-Nocarcinoma" You mean Adenocarcinoma?

-Figure 2 (3) is not readable. Increase the font.

-Authors missed several recent kinds of literature, one example PMID: 32061257. well explains immunotherapy and PDAC.

-So authors must include a Discussion section in the article before the conclusions section. 

e

-Nocarcinoma

Author Response

The manuscript "Where do we stand with immunotherapy for advanced pancreatic ductal adenocarcinoma: a synopsis of clinical outcomes" is well written. I have only minor comments

Starting from the title, what is " Ade-Nocarcinoma" You mean Adenocarcinoma?

Thank you so much for the comment. Please accept our sincere apologies for this technical oversight. For some reason the PC has implied this auto-correction and we did not spot it at the time of first submission. It has been corrected now in the title and checked throughout the whole text too.

Figure 2 (3) is not readable. Increase the font.

Thank you for the valuable comment. We have made two figures from the original one and increased the size of the font. Thank you.

Authors missed several recent kinds of literature, one example PMID: 32061257. well explains immunotherapy and PDAC.

Thank you so much for the comment. Indeed, this manuscript provides a very thorough review of current therapies in PDAC. We have cited this manuscript in the resubmitted version. Thank you.